# Towards Sustainable Road Safety in Saudi Arabia: Exploring Traffic Accident Causes Associated with Driving Behavior Using a Bayesian Belief Network

Muhammad Muhitur Rahman *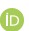, Md Kamrul Islam *, Ammar Al-Shayeb and Md Arifuzzaman

Department of Civil and Environmental Engineering, College of Engineering, King Faisal University, Al-Ahsa 31982, Saudi Arabia; afshayeb@kfu.edu.sa (A.A.-S.); marifuzzaman@kfu.edu.sa (M.A.)
* Correspondence: mrahman@kfu.edu.sa (M.M.R.); maislam@kfu.edu.sa (M.K.I.)

**Abstract:** Understanding the causes and effects of road accidents is critical for developing road and action plans in a country. The causation hypothesis elucidates how accidents occur and may be applied to accident analysis to more precisely anticipate, prevent, and manage road safety programs. Driving behavior is a critical factor to consider when determining the causes of traffic accidents. Inappropriate driving behaviors are a set of acts taken on the roadway that can result in aberrant conditions that may result in road accidents. In this study, using Al-Ahsa city in Saudi Arabia's Eastern Province as a case study, a Bayesian belief network (BBN) model was established by incorporating an expectation–maximization algorithm. The model examines the relationships between indicator variables with a special focus on driving behavior to measure the uncertainty associated with accident outcomes. The BBN was devised to analyze intentional and unintentional driving behaviors that cause different types of accidents and accident severities. The results showed when considering speeding alone, there is a 26% likelihood that collision will occur; this is a 63% increase over the initial estimate. When brake failure was considered in addition to speeding, the likelihood of a collision jumps from 26% to 33%, more than doubling the chance of a collision when compared to the initial value. These findings demonstrated that the BBN model was capable of efficiently investigating the complex linkages between driver behavior and the accident causes that are inherent in road accidents.

**Keywords:** sustainable road safety; driver behavior; multicriteria decision making; Bayesian belief network; causality; sustainable road condition; road safety modeling; Al-Ahsa

## 1. Introduction

A transportation system cannot be sustainable unless it is safe for humans, and human life is the most precious resource. Road safety not only encompasses the steps taken to lower the risk of traffic-related injuries and deaths, but also encompasses the sensation of being safe while on the road and confidence that the user will not be seriously wounded or killed while on it. Safety is now recognized in worldwide environmental policies as being critical to achieving sustainable development and should be a precondition for mobility, particularly in countries where the number of road fatalities remains high. The goal of a sustainable and safe road traffic system is to eliminate road deaths, serious road injuries, and permanent injuries by systematically lowering the underlying risks of the whole traffic system. Human aspects are the key focus: the traffic system may be realistically altered to ensure optimum safety by investigating the behaviors, skills, and limitations of drivers.

Traffic, accidents, and pollution are three issues that are becoming increasingly prominent in urban areas as both the population and vehicle fleet continue to increase. The most detrimental of the three elements outlined above is an accident in or near a city center. According to the World Health Organization (WHO), over 1.3 million people died in road traffic accidents in 2010, while 20 to 50 million people were injured [1]. Between 1975 and 1998, the number of people killed or injured in road traffic accidents increased by around

44% in Malaysia and by over 200% in Colombia and Botswana. The World Health Organization has also forecasted that road accidents will be the sixth largest cause of mortality and the second leading cause of disability in developing nations by the year 2020 if current trends continue [2]. According to a review of 404 accident reports of 14 different types of accidents, the road environment played a role in approximately 14.5% of all accidents [3]. According to the findings of the study by the WHO [1], 30 deaths per 100,000 people were observed in Saudi Arabia in 2007, and 6358 deaths occurred as a result of road accidents. Additionally, according to the WHO report [4], in the Kingdom of Saudi Arabia, traffic accidents are the greatest cause of injury, fatality, and disability, and the cost of treating those who are injured or killed in road traffic accidents was projected to be SAR 652.5 million [2,3]. Officials in Saudi Arabia have revealed that road accidents occur every minute in the country, and the kingdom is considered to be one of the world's top countries when it comes to traffic accidents, with approximately 21 deaths per day, ranking it as the second deadliest country in the Middle East [5,6].

There are many factors that cause traffic accidents. Some of these causes are related to road geometry, and some are related to driver behavior. Driving behavior is one of the significant issues when analyzing the reasons of traffic accidents. A report on the town of Mekelle in northern Ethiopia showed that human risk behavior is behind 96% of accidents [7]. A study in an eastern Mediterranean region showed that 86% of drivers engaged in at least one risky driving behavior while driving [8]. Crash-causing risky driving habits include speeding, ignoring red light signals, sudden lane changes, blocking intersections, not using seat belts, and vehicles turning suddenly [9]. If the effects and extent of inappropriate driving attitudes and behaviors on accident severity and type can be identified, it will be helpful in developing suitable road safety policies that would prevent traffic accidents. Saudi Arabia, similar to many other countries throughout the world, has created tactics and scenarios to help mitigate and resolve traffic disasters when they occur. However, despite the deterrent and awareness measures implemented by the Traffic Department and other concerned departments, which have taken it upon themselves to confront this danger, Saudi Arabia continues to experience a significant problem concerning traffic accidents; therefore, it is essential to analyze the effects of driving behavior on road accidents in the Saudi Arabian context.

*Literature Review*

It is essential to understand the causes and effects of road accidents to adopt appropriate safety strategies and action plans. Some of these causes are reflections of the inappropriate attitudes or behaviors of drivers, such as speeding, suddenly changing lanes, and tailgating [10]. Several studies have attempted to analyze and categorize types of driving behavior. Some researchers divided driving behavior in two broad categories: "cautious" and "aggressive" driving [11]. A driver who does not accelerate, can initiate the breaks of their vehicle unexpectedly, and maintain proper speed is considered to be careful and cautious driver [12]. The Department of Transportation of Pennsylvania described aggressive driving as "the operation of a motor vehicle in a manner that endangers or is likely to endanger persons or property" [10]. Eboli et al. [13] analyzed average speed as well as the 50th and 85th percentile speeds of a road segment of a two-lane Italian rural road and classified driving behavior into three categories: (1) safe, (2) unsafe, and (3) safe but potentially dangerous. In another study conducted by Taubman-Ben-Ari et al. [14], it was suggested that driving behavior be categorized into four groups: (1) careless and reckless driving: this driving behavior is characterized by high speed, illegal maneuvers, and racing to seek thrills when driving; (2) anxious driving: related to ineffective relaxation activities with feelings of tension and alertness when driving; (3) hostile and angry driving: includes drivers with antagonistic attitudes and annoyance as well as anger, and these emotions are expressed by acts such as flashing their headlights at others; and (4) careful and patient behavior: expressed via a good attitude, planning for unforeseen situations in advance, and perfectly following traffic rules and regulations. Yasir Ali et al. [15–17]

conducted simulation studies using the CARRS-Q Advanced Driving Simulator to evaluate various critical driving behaviors across a number of normal driving activities, including car-following, interactions with traffic lights, pedestrian crossings, and lane changes. Their findings implied that drivers who communicate well had a longer time-to-collision when following another vehicle, a longer time-to-collision when approaching a pedestrian, lower deceleration to prevent a crash when changing lanes, and a reduced proclivity to run yellow lights. In general, drivers in a networked environment make more informed (and hence safer) decisions. Using the random parameters Bayesian least absolute shrinkage and selection operator (LASSO) modeling approach, Yue Zhou et al. [18] studied the operational aspects affecting aggressive taxi speeders. Taxi GPS trajectory data in Chengdu, China, was used to extract taxi speeding habits and other operational parameters. The fuzzy C-means clustering approach was used to group taxi speeders into three cohorts based on their hourly speeding frequency and average speeding severity: restrained speeders (RS), moderate speeders (MS), and belligerent speeders (BS). MS and BS are regarded as aggressive taxi speeders compared to RS. With RS as the reference category, several binary logistic models have been built. The Bayesian binary logistic LASSO model with random parameters has been found to capture unobserved heterogeneity and to combat multicollinearity. Mohammad Jalayer et al. [19] applied a random parameter-ordered probit model to identify the attributes of wrong-way driving (WWD) crashes and injuries using 15 years of crash data from the states of Alabama and Illinois. According to the obtained results, factors such as driver age, driver condition, roadway surface conditions, and lighting conditions significantly contribute to the injury severity in WWD crashes. Zhengwu Wang et al. [20] combined a classification tree with a logistic regression model and studied the underlying risk factors for severe injuries in different categories of e-bike users. Three years of e-bike crashes in Hunan province were analyzed by considering risk factors such as rider attributes, opponent vehicle and driver attributes, incorrect riding and driving behaviors, and road and environmental characteristics. Below, Table 1 summarizes the literature related to driving behaviors.

**Table 1.** References on driving behavior.

| References | Driving Behaviors | Comments |
|---|---|---|
| P. McTish and S. Park (2016) [10] | Aggressive driving in Pennsylvania's Delaware Valley region in the USA. | Conducted statistical analysis among aggressive crash features (e.g., type, severity level), roadway features (operation and geometric), and driver behavior. |
| C. Wang et al. (2014) [11] | Different driving styles, vision sensors, radar, GPS, and vehicle CAN bus data capture systems were installed in a small passenger car, and a real road driving test was carried out. | Used the fuzzy evaluation method to categorize driving behavior. |
| G. Miller and O. Taubman-Ben-Ari, (2010) [12] | Studied the risky behavior of young novice drivers. | Analyzed the contribution of parental driving styles and personal characteristics on the behavior of young drivers. |
| L. Eboli et al. (2017) [13] | Classified driving behavior into three categories: (1) safe, (2) unsafe, and (3) safe, with potentially dangerous behavior based on speed analysis. | Conducted a survey to collect experimental speeds in a real situation in an Italian rural two-lane road. |

**Table 1.** *Cont.*

| References | Driving Behaviors | Comments |
|---|---|---|
| O. Taubman-Ben-Ari et al. (2004) [14] | Developed a self-report scale assessing four broad domains of driving styles—the multidimensional driving style inventory (MDSI). | Applied factor analysis that revealed eight main factors, with each one representing a specific driving style—dissociative, anxious, risky, angry, high-velocity, distress reduction, patient, and careful. |
| Y. Ali et al. (2020) [15–17] | Evaluated various critical driving behaviors across a number of normal driving activities, including car following, interactions with traffic lights, pedestrian crossings, and lane changes. | Conducted simulation studies using the CARRS-Q Advanced Driving Simulator. |
| Yue Zhou et al. (2021) [18] | Operational aspects affecting aggressive taxi speeders. | Applied the random parameter Bayesian least absolute shrinkage and selection operator (LASSO) modeling approach on taxi GPS trajectory data from Chengdu, China. |
| Mohammad Jalayer et al. (2018) [19] | Identified the attributes of wrong-way driving (WWD) crashes and injuries. | Applied random parameter-ordered probit model using 15 years of crash data from Alabama and Illinois, USA. |
| Zhengwu Wang et al. (2021) [20] | Studied the underlying risk factors for severe injuries in different categories of e-bike users. | Combined a classification tree with a logistic regression model using e-bike crash data from Hunan province, China. |

The causality hypothesis can be applied to accident analysis to anticipate, prevent, and manage road safety initiatives on a more precise level in order to explain how driver behavior leads to accidents. Researchers from all around the world have undertaken numerous studies on the causality of road accidents through the use of a variety of data sets, locations, sample sizes, and factors as well as analytical models to determine the causes of accidents. As an example, references [21–23] provided aggregate models in which an accident frequency analysis and $\chi^2$ Test [21,24] were devised as key tools. Lord et al. [25] proposed a non-negative, discrete disaggregated model in which they assumed from experience that the rate of accidents follows a Poisson distribution and applied a Poisson distribution to observe the influence of risk factors on the rate of accidents. Researchers [26–28] have extensively employed the negative binomial regression model for road safety analysis; nevertheless, it has been discovered that the real-world scenarios do not always correspond to the assumption of equal mean and variance required for the Poisson distribution, as stated in different studies. Furthermore, when applying the Poisson regression model and negative binomial regression to longitudinal data samples, there is a substantial danger of obtaining a skewed estimate, if not completely wrong results. These models are based on invariant parameters that do not account for temporal variability. There are some recent modelling techniques that analyze traffic accidents in real time and that update the model parameters recursively and react to abrupt trend changes [29–31].

Researchers have gradually advanced from developing aggregated models to developing complex disaggregated models; aggregate modeling entails straightforward descriptive analysis, whereas disaggregate modeling entails complex multivariate analysis. However, there is a persistent lack of understanding of the circumstances, meaning that accident-causing elements have not been completely explored. The existing literature is deficient in that the majority of analyses and models are isolated, single factor, or case-specific and fail to present, correlate, and explain the underlying processes and complex multidimensional relationships between accident causes, occurrences, and consequences. Although some scholars have attempted to address these issues [32–37], the theoretical and empirical foundations for accident mechanisms have not been established systematically.

This paper aims to examine the interrelationship between different factors caused by inappropriate driving behaviors and the sequential effects that cause road accidents. Causal probability can describe such a relationship by applying Bayesian belief network (BBN) analysis. Since BBN has stronger objectivity and explanatory ability, it can better illustrate a relationship between different accident variables in complex systems compared to regression and other multi-criteria analysis methods [38]. BBN is one of the most prominent models in probabilistic modeling [16], in which a given collection of random variables is presented as a joint probability distribution for a set of observations. In terms of learning from data, BNNs may be divided into two linked steps: (i) network structure learning and (ii) parameter learning [17,18]. When it comes to learning the structure of a network, there are two approaches: one such approach is the constraint-based approach. Based on the results of a large number of independent tests on the database, this approach is followed by the construction of a BBN following the results of the tests. An important example of this strategy is the parents and children (PC) algorithm [19]. The other method is referred to as a score-based strategy in most cases. To evaluate a given BBN model, this methodology employs some edge-based scoring and searching algorithms, which are described in detail by reference [17]. Moreover, the Bayesian network method has been applied by many researchers in the modeling of road safety [37,39–42]. There are also several studies where BBN is applied to analyze maritime accidents [43–46].

BBN has been applied to analyze traffic accidents in different countries. For example, Zou and Yue [47] studied accident causation in Australia, Karimnezhad and Moradi [48] studied the causes of accidents in Iran, Deublein et al. [49] studied the causes of accidents in Switzerland, and Zamzuri et al. [50] studied the causes of accidents in Malaysia. However, to the authors' knowledge, no BBN model has been proposed for road safety modeling in Saudi Arabia. This article proposes a Bayesian belief network for the causation analysis of road accidents using the Al-Ahsa region of Saudi Arabia as a case study with the most recent data available. The specific objective of the present study is to evaluate the effect of driving behavior on accident type and severity in the case study area.

This paper is arranged as follows: In Section 2, the study area and data sources are described. Section 3 presents the BBN model. Section 4 describes model development along with discretized variables. The application of the model with calibration and validation is discussed in Section 5. Finally, Section 6 completes the paper with a summary and conclusions.

## 2. Case Study Area and Data Sources

Al-Ahsa city in the Eastern Province of Saudi Arabia is the case study area for this study (Figure 1). Al-Ahsa was selected in this study because of its high accident rate, which has been reported in recent literature [51]. It was observed that within the Eastern Province between 2009 and 2016, 31.9% of accidents were recorded in Al-Ahsa, followed by Dammam, Hafr Al-Batin, Jubail, Qatif, Dhahran, and Khobar (Figure 2). Accidents in other cities were recorded to be below 5%. This certainly indicates that Al-Ahsa city is the most vulnerable area for traffic accidents among all of the other cities in the Eastern Province. It should be mentioned that recently, Al-Ahsa has been nominated a UNESCO-listed heritage site in Saudi Arabia and has received an award from the Guinness Book of World Records, which has provided Al-Ahsa with significant potential to be an international tourist attraction. However, the high accident rate in this city may pose risk to that potential and requires appropriate action.

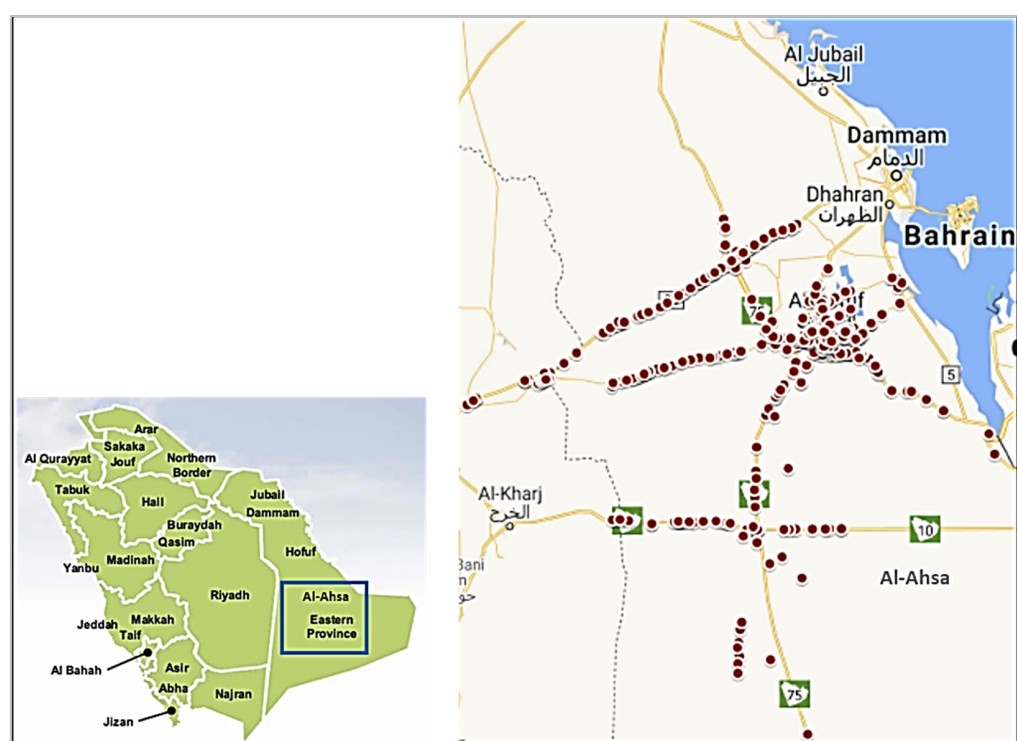

**Figure 1.** Study area and accident locations between 2015 and 2018 in the study area of Al-Ahsa, Eastern Province.

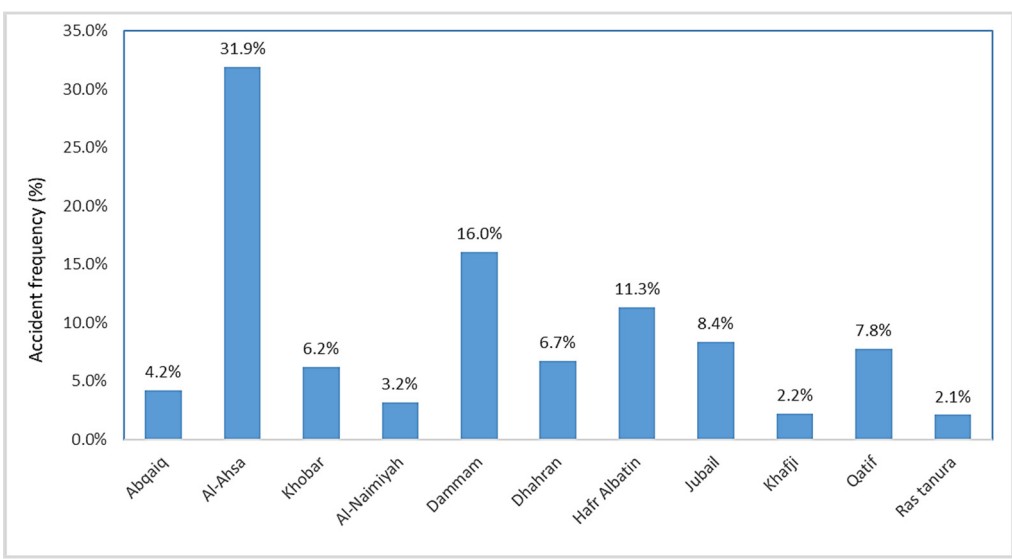

**Figure 2.** Distribution of traffic accidents in the Eastern Province of Saudi Arabia (adopted from [51]).

The accident data for Al-Ahsa used in this study were provided by the Traffic Police Department, Dammam, for the period from October 2014 to May 2018. During this time, 3994 accidents were recorded in Al-Ahsa. An overall observation of the data set indicates that collisions between vehicles was the most predominant type of accident, causing around 8% and 30% of fatal and injury accidents, respectively (Figure 3). Vehicle overturning was another primary accident type, resulting in 6.5% and 12.5% of fatal and injury accidents, respectively. Furthermore, pedestrian vulnerability was reflected in 12.5% of injury accidents and 2% of fatal accidents. Some accident types such as hitting road fences, motorcycles, parked vehicles, and fixed objects also show the potential threat of injury, whereas fatal accidents rates were relatively low in these categories. It should be mentioned that statistical

data from 2015 to 2018 were used in this analysis for the construction of the Bayesian belief network model (Table 2).

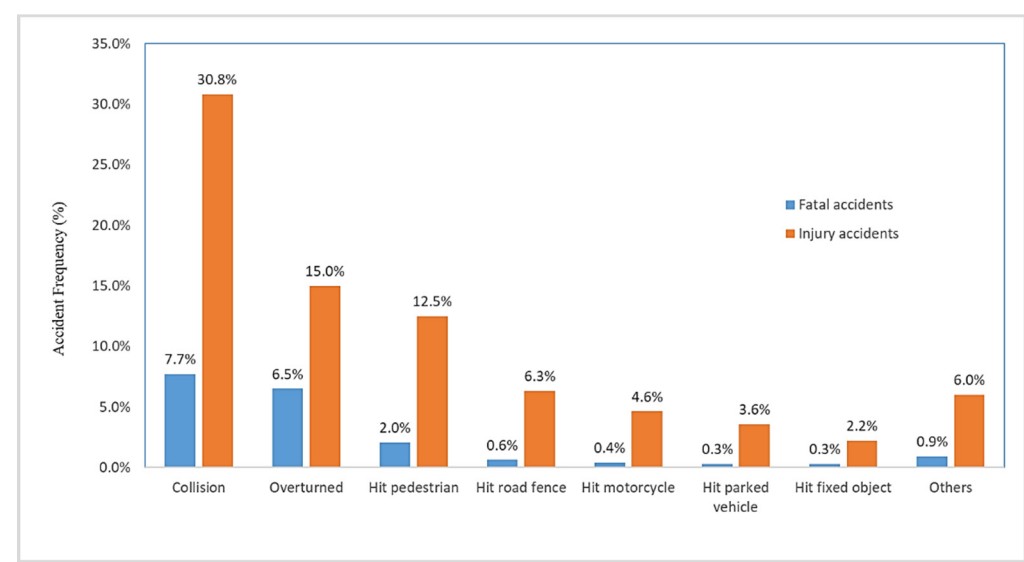

**Figure 3.** Distribution of accident frequency with types of accidents in the study area.

**Table 2.** Description of discretized variables used in the BBN.

| Name of the Variable | No. | Discretization | Frequency | % |
|---|---|---|---|---|
| Unintentional Driver Behavior | 1 | Falling asleep | 2 | 67.0% |
| | 2 | Exhaustion | 1 | 33.0% |
| Intentional Driver Behavior | 1 | Driver distraction | 230 | 5.8% |
| | 2 | Speeding | 725 | 18.4% |
| | 3 | Red light violation | 13 | 0.3% |
| | 4 | Not stopping at STOP sign | 7 | 0.2% |
| | 5 | Driving opposite to traffic | 10 | 0.3% |
| | 6 | Illegal overtaking | 22 | 0.6% |
| | 7 | Not giving way | 615 | 15.6% |
| | 8 | Drifting | 5 | 0.1% |
| | 9 | Sudden lane changes | 1918 | 48.7% |
| | 10 | Insufficient safe distance | 256 | 6.5% |
| | 11 | Other | 138 | 3.5% |
| Vehicle Condition | 1 | Faulty tires | 46 | 95.8% |
| | 2 | Faulty breaks | 1 | 2.1% |
| | 3 | Faulty electric | 1 | 2.1% |
| Seasons | 1 | Winter (December–February) | 1014 | 25.4% |
| | 2 | Spring (March–May) | 1127 | 28.2% |
| | 3 | Summer (June–August) | 960 | 24.0% |
| | 4 | Autumn (September–November) | 893 | 22.4% |
| Accident tType | 1 | Collision (both head-on and rear-end) | 1542 | 38.5% |
| | 2 | Overturned | 862 | 21.5% |
| | 3 | Hit pedestrian | 582 | 14.5% |
| | 4 | Hit road fence | 278 | 6.9% |
| | 5 | Hit motorcycle | 201 | 5.0% |
| | 6 | Hit parked vehicle | 154 | 3.8% |
| | 7 | Hit fixed object | 100 | 2.5% |
| | 8 | Other | 275 | 6.9% |

**Table 2.** *Cont.*

| Name of the Variable | No. | Discretization | Frequency | % |
|---|---|---|---|---|
| Accident severity | 1 | Fatal | 756 | 18.9% |
| | 2 | Injuries | 3238 | 81.1% |

## 3. BBN Model

The BBN is employed as the foundation for the evaluation framework in this research. The BBN algorithm performs adequately, even in the presence of insufficient data [52]. A network diagram depicts the causal relationships of an event or variable in graphical form. Rather than considering occurrences, this study considers the network diagram for various factors. The model was founded on Bayes' theorem, as stated in reference [53]:

$$P(A|B) = P(A)\left[\frac{P(B|A)}{P(B)}\right] \tag{1}$$

where $P(A|B)$ = the posterior probability; $P(A)$ and $P(B)$ = the prior probability; and $P(B|A)$ = the likelihood of $B$.

BBN parameters have several states. $A = a_1, \ldots a_n$ and $B = b_1, \ldots b_n$ denote the states coupled with variables $A$ and $B$. The probabilities of variables $A$ and $B$ can be represented as follows [54]:

$$\begin{aligned}
P(A) &= (x_1, \ldots \ldots, x_n); x_i \geq 0); \sum_{i=1}^{n} x_i = x_1 + \ldots + x_n = 1) \\
P(B) &= (y_1, \ldots \ldots, y_m); y_j \geq 0); \sum_{j=1}^{m} y_j = y_1 + \ldots + y_m = 1)
\end{aligned} \tag{2}$$

Here, $x_i$ is the probability of $A$ being in the state $a_i$, and $y_j$ is the probability of $B$ being in the state $b_j$.

Essentially, Bayes' theorem (Equation (1)) is predicated on the conditional probability (CP) of variables. The term $P(A | B)$ contains $n \times m$ conditional probabilities that specify the likelihood of obtaining $a_i$ when given $b_j$. This indicates that conditional probability is a table of $n \times m$ probabilities, one for each configuration of the variable states. This table is referred to as the conditional probability table (CPT).

BBN is constructed by identifying causative linkages among variables that are visually represented by nodes that are connected by arrows. *Children* are the variables with inward-facing arrows, while *parents* are variables with outward-facing arrows. Conditional probabilities are assigned to the child nodes, while marginal probabilities are assigned to parent nodes. To illustrate CPT, if a child node $A$ has several parent nodes $B_1, B_2 \ldots, B_n$, a CPT is linked to the node $P(A|B_1, B_2 \ldots, B_n)$ a CPT is linked to the node $P(A | B_1, B_2 \ldots, B_n)$.

## 4. Proposed Model Development

Figure 4 depicts the procedures involved in developing and applying the BBN model. The model was built and used in six steps: (i) parent–child node identification; (ii) the collection of accident data from relevant agencies and from related literature; (iii) data analysis to find the prior beliefs of parent nodes; (iv) framework construction in Hugin-Expert™ software to generate CPT using the EM (estimation–maximization) algorithm; (v) testing the developed model by comparing the modeled results with the observed results during the study period; and (vi) the application of the developed model to evaluate the causal relationship between driving behavior and accident type.

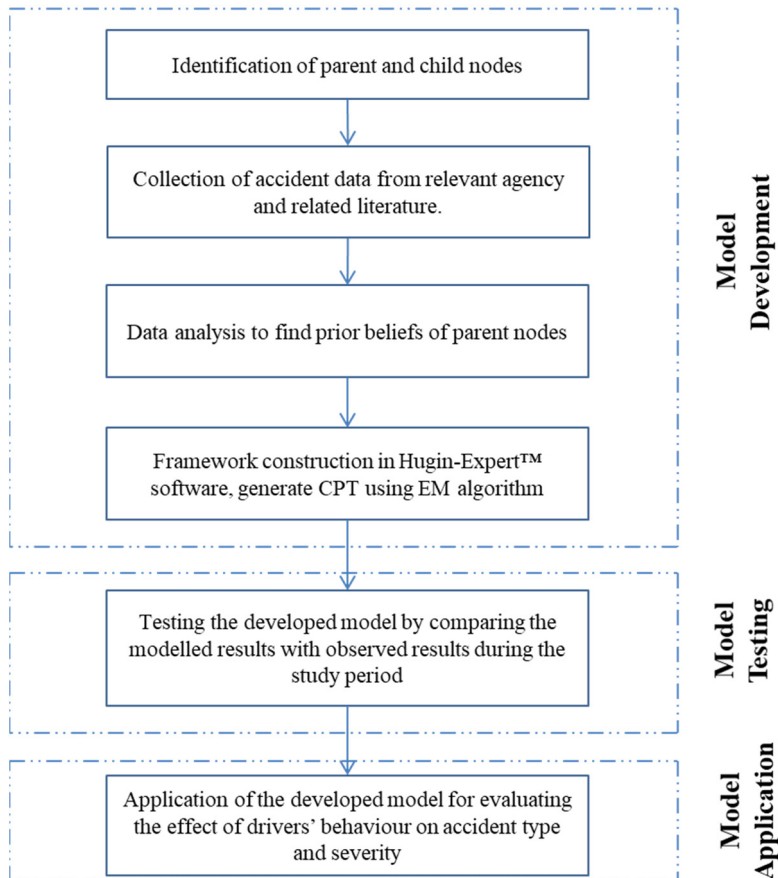

**Figure 4.** Steps for Bayesian belief network (BBN) analysis.

In this study, the CPT was generated using the EM algorithm implemented in the Hugin-Expert™ [55]. The EM algorithm is an approach that may be used to estimate the parameters of a conditional probability table given a set of cases [48]. The method is widely used when the data set has missing or incomplete data [48,49]. The algorithm has two steps, namely the expectation step (E-step) and the maximization step (M-step) [56], and can be summarized as below:

Let us assume $\theta$ to be the parameters in a Bayesian network $\mathcal{N} = (\mathcal{G}, \mathcal{P})$, where $G$ = the directed graph (DAG) and $\mathcal{P}$ = the probability, and $\theta_{ijk} = P(X_i = k | pa(X_i) = j)$.

The algorithm is built on the probable value of the log-likelihood function and can be written as:

$$Q(\theta^*|\theta) = \mathbb{E}_\theta\{logP(X|\theta^*)|D\}$$

where $P$ = the density function of $X$; $D$ = the observed data = $g(X)$.

For an initial value of the parameters $\theta$, the E-step computes the value of $Q$ in terms of $\theta$, and the M-step maximizes $Q$ in $\theta^*$. Both processes will be iterated alternately until a certain end requirement is met. The log-likelihood function $l(\theta|D)$ of the parameters $\theta$ provided by the data $D$, and DAG $G$, can be represented as follows:

$$l(\theta|D) = \sum_{i=1}^{N} \log P(c_i|\theta)$$

In a BBN, the E-step will compute expected counts for the family $fa(X_i)$ and parent $pa(X_i)$ configurations of each node $X_i$ under $\theta$:

$$n^*(Y) = \mathbb{E}_\theta\{n(Y)|D\}$$

where $Y$ is either $pa(X_i) = j$ or $(X_i) = k$, and $pa(X_i) = j$. The M-step then computes new estimates of $\theta^*_{ijk}$ from the expected counts under $\theta_{ijk}$:

$$\theta^*_{ijk} = \frac{n^*(X_i = k,\ pa(X_i) = j)}{n^*(pa(X_i) = j)}$$

The E-step and M-step are iterated until the convergence of $l(\theta)$.

*Variables (Parent and Child)*

The notion behind the identification and classification of the parent and child nodes for BBN construction in this research was guided by the in-depth discussion on the possible causes of accidents reported by previous studies [51,57,58]. Subsequently, six main variables, i.e., *driver behavior* (intentional and unintentional), *vehicle condition*, *season*, *accident type*, and *accident severity*, were nominated from the data sets, as presented in Table 2. Among the variables, *driver behavior*, *vehicle condition*, and *season* were parent nodes, whereas accident type and accident severity were child nodes. The suggested BBN (Figure 5) is a directed graph and is widely regarded as the most appropriate representation for causative interactions between parent and child variables [59,60]. By definition, a directed graph is acyclic when there is no cycle existing in the spotlight pathway among the parent and child nodes in the model structure.

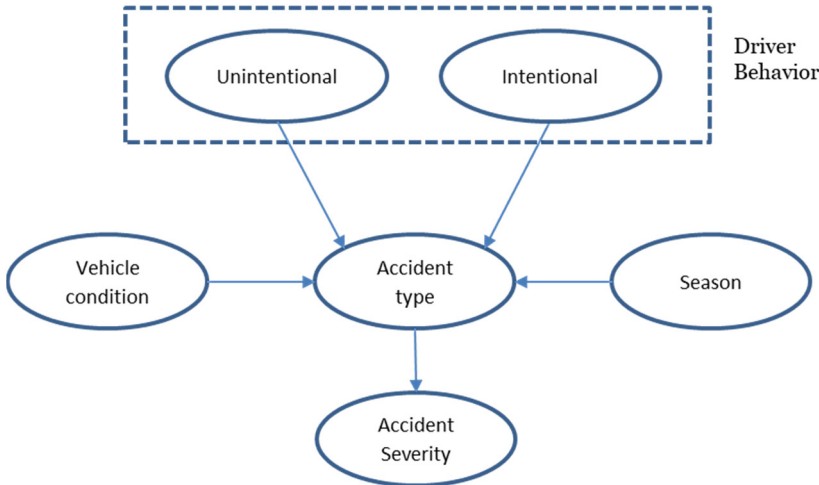

**Figure 5.** Proposed BBN model for accident analysis.

The predominant causes of road accidents in Al-Ahsa, as established by the observed data (Table 3), mainly fall under intentional driver behavior, namely, distractions, speeding, sudden lane changes, not giving way, insufficient safe distance, violating signals and signage, illegal overtaking and driving, and drifting. A similar observation was also noted by reference [51] for the Eastern Province of Saudi Arabia. "Other" intentional behaviors include getting out of the vehicle before stopping, getting in a vehicle before stopping, hanging from the outside of a vehicle, sitting on top of a vehicle, sitting on the trunk of a vehicle, and violating pedestrian signage, constituting 3.5% of road accidents in Al-Ahsa. In the case of *unintentional behavior*, although the frequency of the occurrence is low, considering the importance of the phenomena, two causes were considered from this variable in the BBN network. The parent nodes, *vehicle condition* and *season*, were introduced into the network to check whether vehicle fitness and weather conditions are probable causes of accidents and road safety. It should be noted that, although important, the variable "not using seat belts" is not included in the model due to a lack of data by the research team. The accident type "property damage only (PDO)" is also not modelled for the same reason. Bayesian networks are capable of processing both continuous and discrete information. Due to the clear discrete nature of the classification results for traffic

accident variables, discrete variables are used for the BBN framework for road accidents. Before structure learning, the road accident variable must be discretized. Table 2 contains the discretized variables used in the network.

**Table 3.** Sensitivity analysis results for the target node "Accident Type".

| Table | Parent Node | Unintentional Driver Behavior | Intentional Driver Behavior | Vehicle Condition | Season |
|---|---|---|---|---|---|
| All states | Max | 0.05000 | 0.29000 | 0.07000 | 0.03000 |
| | Min | −0.00743 | −0.00844 | −0.00844 | −0.00957 |
| Collision | Max | 0.05000 | 0.29000 | 0.05000 | 0.01000 |
| | Min | −0.05000 | −0.09000 | −0.02000 | −0.02000 |
| Overturned | Max | 0.02000 | 0.23000 | 0.07000 | 0.02000 |
| | Min | −0.02000 | −0.00844 | −0.06000 | −0.03000 |
| Hit pedestrian | Max | 0.00267 | 0.05000 | 0.04000 | 0.01000 |
| | Min | −0.00267 | −0.00541 | −0.05000 | −0.0024 |
| Hit road fence | Max | 0.02000 | 0.14000 | 0.04000 | 0.01000 |
| | Min | −0.02000 | −0.00399 | −0.00647 | −0.00925 |
| Hit motorcycle | Max | 0.00743 | 0.03000 | 0.01000 | 0.02000 |
| | Min | −0.00743 | −0.00620 | −0.00844 | −0.00802 |
| Hit parked vehicle | Max | 0.02000 | 0.05000 | 0.04000 | 0.01000 |
| | Min | −0.02000 | −0.08000 | −0.07000 | −0.00957 |
| Hit fixed object | Max | 0.01000 | 0.04000 | 0.06000 | 0.00539 |
| | Min | −0.01000 | −0.08000 | −0.06000 | −0.00790 |
| Other | Max | 0.00253 | 0.20000 | 0.05000 | 0.03000 |
| | Min | −0.00253 | −0.07000 | −0.02000 | −0.02000 |

## 5. Results and Discussion

The developed model was implemented utilizing the Hugin-Expert$^{TM}$ software. Hugin was used to obtain the prior probability distributions for all of the nodes, which are shown in Figure 6. Each box in the picture characterizes a three-column variable. The left two columns illustrate and quantitatively reflect the probability distribution. The other column depicts the probability distribution's various states. A careful analysis of Figure 6 reveals that the parent node distribution is very similar to the raw data shown in the prior sections, confirming that the parent–child ties are acceptable. The Bayes' equation was used to determine how the evidence entered into the child node affected the state probabilities of the parent variables. More specifically, the child nodes were given a known probability, which altered the probability in the parent nodes. This process is referred to as *backward propagation* [60]. Similarly, inserting a known probability into the parent variables modifies the probability in the child variables, which is referred to as *forward propagation*.

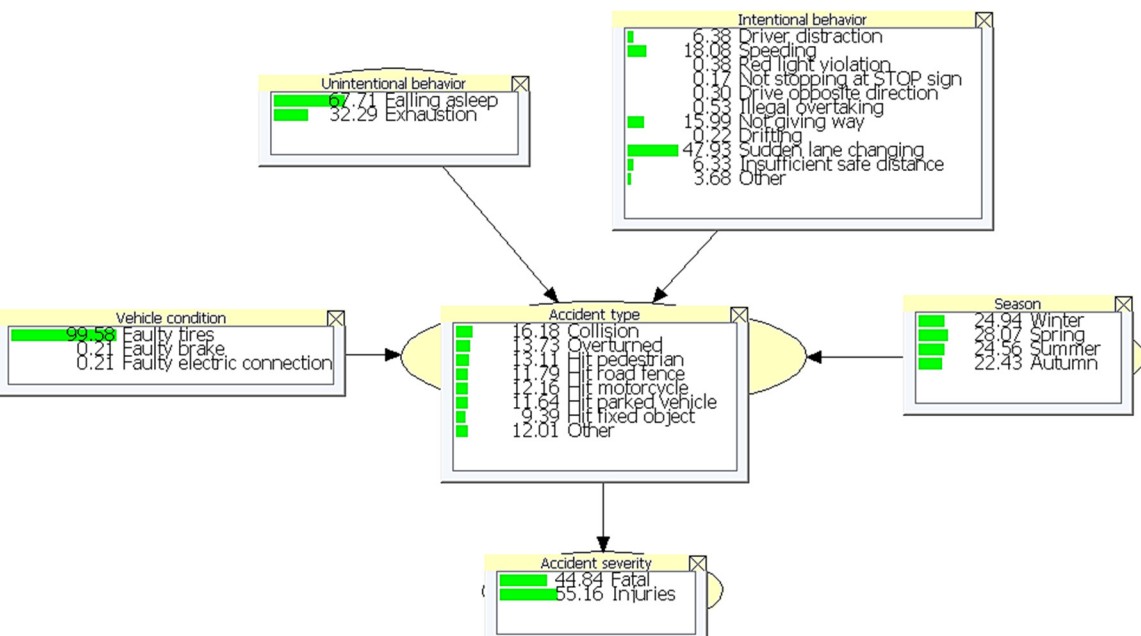

**Figure 6.** Status of probability in variables after parameter learning in Hugin-Expert™ (extracted from Hugin-Expert output).

### 5.1. Model Calibration and Validation

It is required to perform sensitivity analysis to determine the important elements influencing the concerned node. After that, the concerned node can be utilized as evidence to fit and forecast using models that include these important factors.

### 5.1.1. Sensitivity Analysis

Sensitivity analysis allows us to identify which of the parameter values from the CPT or probability (node state) are related to the value of the parameter in question [61]. Due to the structure of the evidence (there is no evidence downstream of the concerned variable), the functional connection is linear [62]. Hugin-Expert's sensitivity analysis feature was used to determine the factors with higher values when evaluating road incidents. The target node was selected in Hugin-Expert, and then the influence on the target node was analyzed. Table 3 shows the results of a sensitivity analysis of the "Accident Type" node. It is evident from the table that out of all of the state values, *Intentional Driver Behavior* is more sensitive to the Accident Type compared to other nodes and hence needs further investigation.

### 5.1.2. Model Fitting

The posterior probabilities for "Unintentional Behavior", "Intentional Behavior", "Vehicle Condition", and "Season" produced by the BBN were compared to real observations from 2015 to 2018 (Figure 7). Due to the volume of data, the "Collision" state from the "Accident Type" node was used to illustrate the concept. As seen from the figure, there is excellent agreement between the modeled and observed values, which is further supported by low MAE and PBIAS values.

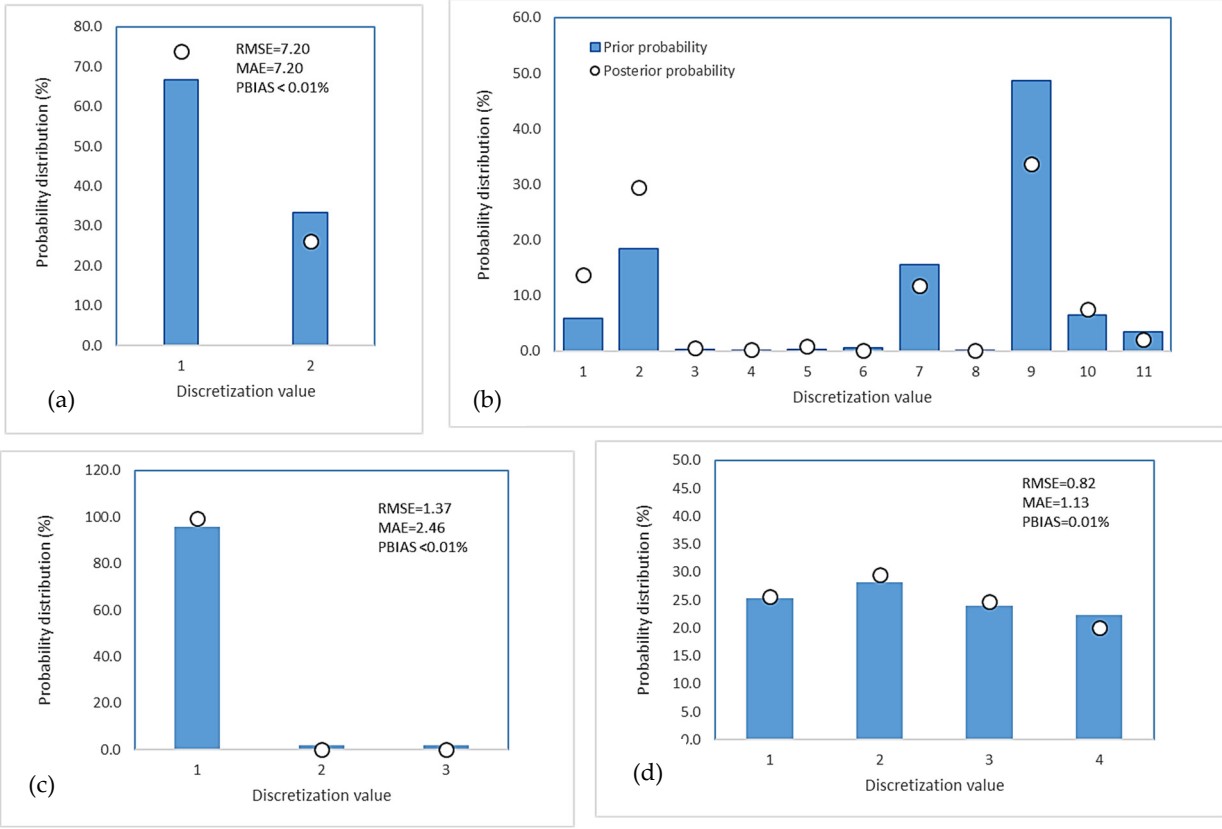

**Figure 7.** Comparing the posterior and real probabilities of different parent nodes when "Collision" is used as the evidence; (**a**) unintentional behavior, (**b**) intentional behavior, (**c**) vehicle condition, and (**d**) season.

*5.2. BBN Model Application to Evaluate Likelihood of Accident Type Due to Drivers' Behavior*

The likelihood of an accident shown in Figure 8 is established on the assumption that a driver is "speeding" in Al-Hofuf city center, Al-Ahsa. When evidence was entered to account for 100% (red color, in Figure 8) in this state in Hugin-Expert, the probabilities of the other variables changed following the relationship entered in the network structure. It can be seen that due to speeding, there is a 26.3% chance that the accident type would be a "Collision". This is an increase of 63% compared to the initial value. Speeding was found to be a major cause of road accidents in Riyadh [52,58] and in the Qassim region [57,63] of Saudi Arabia, as well as in Kuwait [63], and this was found by other researchers also [64,65]. According to Mohamed and Bromfield [57], the negative attitudes that drivers have towards road safety and aggressiveness are major causes of speeding-related crashes in Saudi Arabia.

To observe the changes in the likelihood in all states of "Accident Type" and "Accident Severity", similar forward propagation exercises were carried out for evidence entered for the different states of the "Unintentional Behavior" and "Intentional Behavior" variables, as shown in Table 4. Table 4 shows a significant increase in the accident risk of all accident type states. It can be seen that driving in the opposite direction and not maintaining sufficient safe distance, for example, result in an increased accident risk of "Collision" of 45% and 19.33 % r, respectively. These observations are supported by studies in Kuwait [63] and in Shenzhen, China [66]. Similarly, accidents due to "Drifting" would increase to 10%. Ramisetty-Mikler and Almakadma [67] conducted a survey among adolescent motorists in Riyadh and showed that "Drifting" is an act of adventure for them even though it is known as a hazardous behavior.

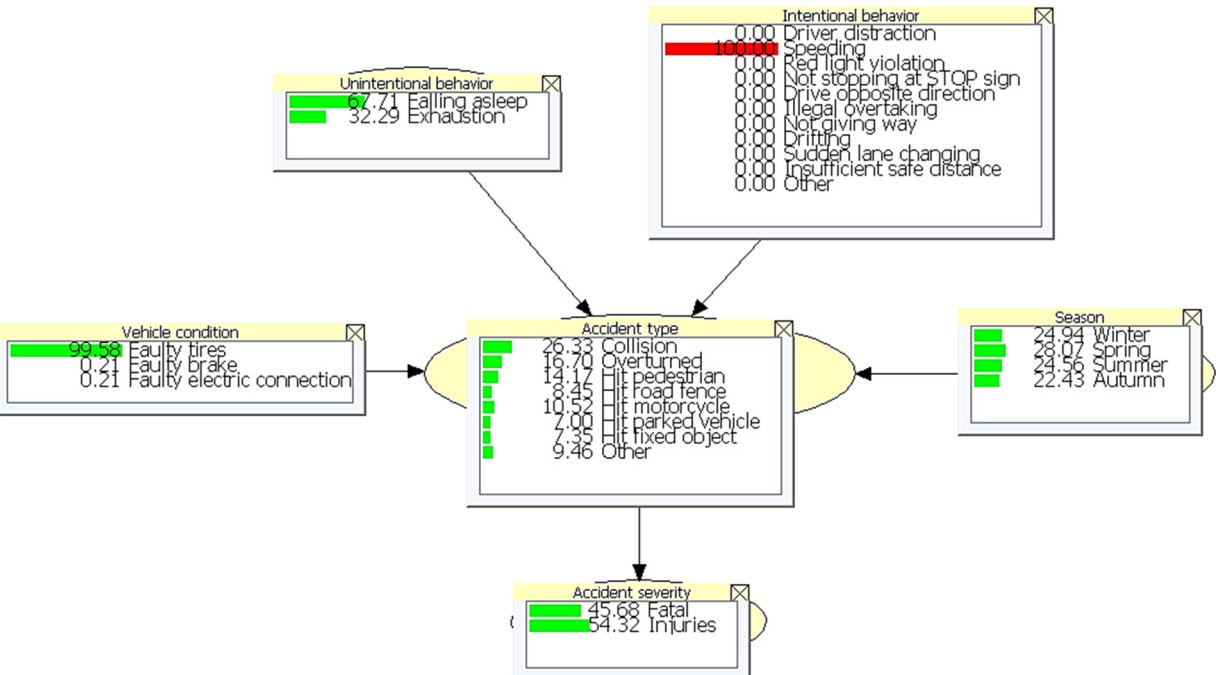

**Figure 8.** Likelihood of an accident given the evidence state of "Speeding" (extracted from Hugin-Expert output).

Further analysis was conducted to see the consequences of "Speeding" and "Faulty brake" at the same time. The result is shown in Figure 9 and shows that the chance of a collision increases from 26.3% to 33.34%, more than doubling the chance of a collision compared to the initial value. All of these observations prove that the model can be employed to predict the accident type given driver behavior and the vehicle conditions.

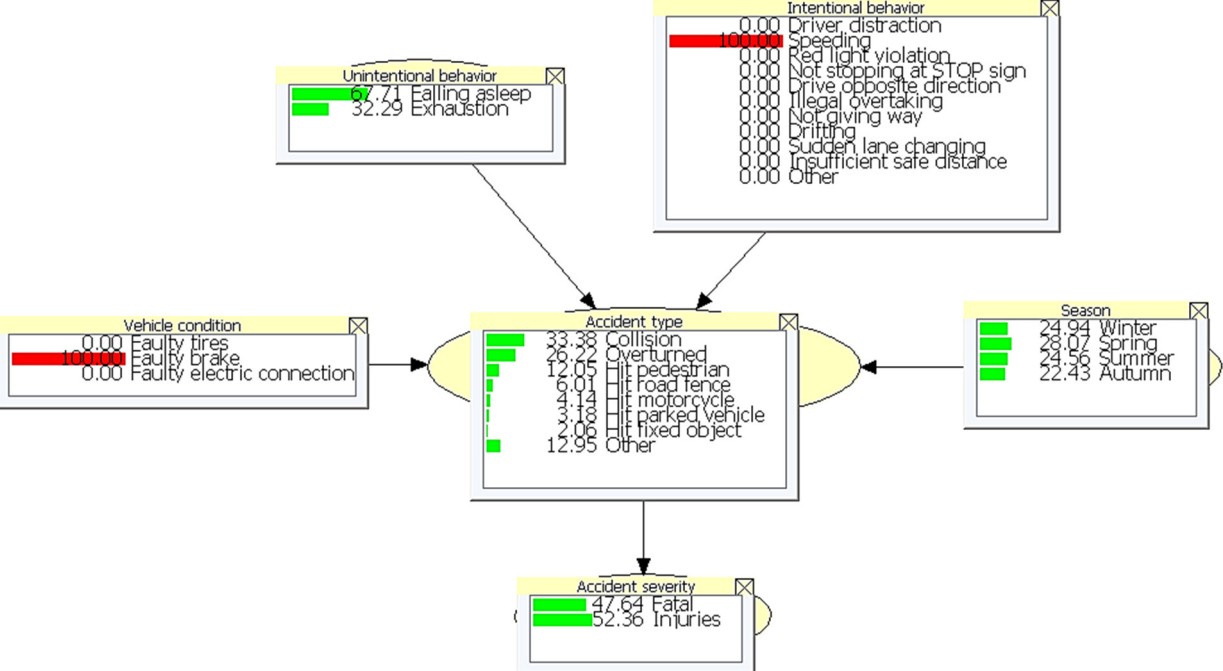

**Figure 9.** Likelihood of accident given the evidence states "Speeding" and "Faulty brake" (extracted from Hugin-Expert output).

**Table 4.** Posterior probability in the states of the child nodes for evidence entered in different states of the "Unintentional Behavior" and "Intentional Behavior" variables.

| Discretization Description | Prior Probability | Evidence in Unintentional Behavior is 100% | | | | | Evidence in Intentional Behavior is 100% | | | | | | | |
| | | Falling Asleep | Exhaustion | Driver Distraction | Speeding | Red Light Violation | Not Stopping at STOP Sign | Drive Opposite Direction | Illegal Overtaking | Not Giving Way | Drifting | Sudden Lane Change | Insufficient Safe Distance | Other |
| | | Posterior Probability | | | | | | | | | | | | |
| Collision | 38.5 | 17.65 | 13.09 | 34.81 | 26.31 | 22.52 | 23.36 | 45.08 | 2.72 | 11.83 | 10.12 | 11.37 | 19.33 | 8.92 |
| Overturned | 21.5 | 14.23 | 12.66 | 19.55 | 16.7 | 25.39 | 11.05 | 6.63 | 11.25 | 13.02 | 37.1 | 11.57 | 16.98 | 13.05 |
| Hit pedestrian | 14.5 | 13.02 | 13.29 | 15.19 | 14.16 | 9.41 | 13.75 | 7.05 | 6.4 | 12.65 | 11.62 | 12.93 | 8.9 | 17.73 |
| Hit road fence | 6.9 | 11.22 | 12.99 | 6.49 | 8.46 | 6.14 | 25.46 | 4.33 | 18.34 | 12.27 | 7.64 | 13.44 | 11.42 | 14.36 |
| Hit motorcycle | 5.0 | 11.93 | 12.67 | 7.15 | 10.54 | 7.25 | 2.52 | 0.98 | 4.42 | 12.75 | 11.63 | 13.6 | 11.59 | 11.78 |
| Hit parked vehicle | 3.8 | 10.85 | 13.29 | 4.77 | 7 | 9.16 | 16.72 | 4.01 | 16.57 | 13.27 | 7.74 | 13.26 | 15.14 | 12.23 |
| Hit electric post (fixed object) | 2.5 | 9.01 | 10.17 | 5.2 | 7.36 | 7.98 | 1.65 | 13.8 | 7.91 | 10.5 | 1.16 | 10.67 | 6.15 | 11.46 |
| Other | 6.9 | 12.10 | 11.84 | 6.84 | 9.47 | 12.16 | 5.50 | 18.12 | 32.38 | 13.71 | 12.99 | 13.17 | 10.49 | 10.47 |
| Fatal | 18.9 | 44.97 | 44.56 | 46.68 | 45.68 | 46.07 | 45.74 | 46.66 | 46.09 | 44.61 | 46.34 | 44.34 | 44.36 | 45.24 |
| Serious injury | 81.1 | 55.03 | 55.44 | 53.32 | 54.32 | 53.93 | 54.26 | 53.34 | 53.91 | 55.39 | 53.66 | 55.66 | 55.64 | 54.76 |

Fault diagnosis is another significant application of Bayesian networks. With the Bayesian network's bidirectional reasoning technology, computational analysis has become easier and more versatile [47]. *Backward propagation* was conducted using the state "Collision" in the node "Accident Type" as an example of causal inference. As seen in Figure 10, once the evidence has been entered (100% probability, in red color), the probability of "Speeding" in "Intentional Behavior" increases greatly from 18.1% to 29.4%. However, the probability of "sudden lane change" decreased. The results indicate that, with the lack of further evidence, it seems most likely that a "Collision" was caused by "speeding" and that "sudden lane change" had less of an impact on the consequence of a collision. However, our observation about abrupt lane changes contradicts a study conducted by reference [68]; the authors determined that when the distance between two vehicles is very small, there is a high probability of a rear-end accident if the leading vehicle uses emergency braking. Similar exercises were carried out for all of the states in the "accident type" variable to observe the change in probability in the states in all four parent nodes, as shown in Table 5.

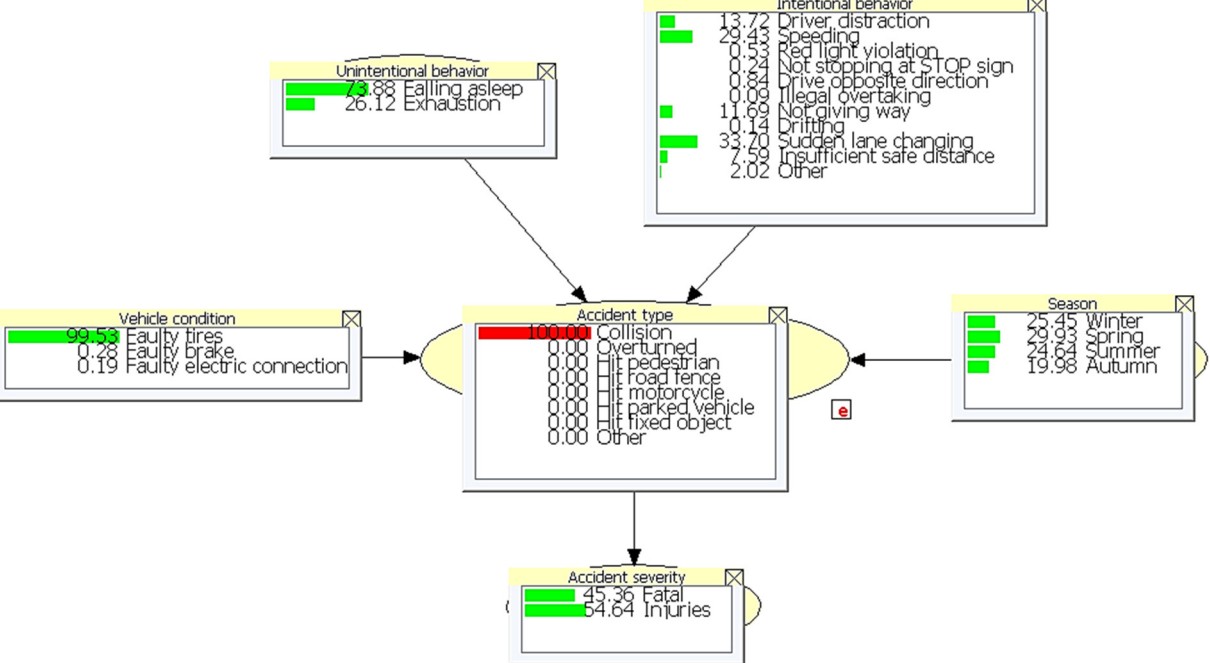

**Figure 10.** Probability distribution in the parent nodes given the evidence state "Collision" (extracted from Hugin-Expert output).

Based on the results presented in the preceding sections, it is imperative that measures be taken to tackle driver behavior as well as driver education and awareness as well as to adopt the latest technological supports to increase road safety in the case study area. Road accidents are the cause of a large number of deaths each year in Saudi Arabia despite the severe penalties imposed by the General Traffic Department for violators of the law [69]. Due to a large number of variables and contributing factors, traffic accidents are one of the most significant problems that plague society and its stakeholders. Because it is a behavioral problem of a complicated nature, it affects a wide range of stakeholders. The prevention of traffic accidents, including their devastating effects, begins with planning and scientific tactics that have been thoroughly thought through. Several studies have offered solutions to reduce and manage the global burden of road accidents [70–75], and the following suggested scenarios provide keys to reducing accident-related injuries and fatalities in Al-Ahsa.

**Table 5.** Posterior probability in the states of all of the parent nodes for evidence entered in different states of the "Accident Type" variable.

| Variable Name | Discretization Description | Evidence in Accident Type is 100% | | | | | | |
|---|---|---|---|---|---|---|---|---|
| | | Collision | Overturned | Hit Pedestrian | Hit Road Fence | Hit Motorcycle | Hit Parked Vehicle | Hit Fixed Object |
| | | Posterior Probability | Posterior Probability | Posterior Probability | Posterior Probability | Posterior Probability | Posterior Probability | Posterior Probability |
| Unintentional Driver Behavior | Falling asleep | 73.87 | 70.21 | 67.26 | 64.43 | 66.37 | 63.13 | 65 |
| | Exhaustion | 26.13 | 29.79 | 32.74 | 35.57 | 33.63 | 36.87 | 35 |
| Intentional Driver Behavior | Driver distraction | 13.74 | 9.09 | 7.4 | 3.52 | 3.75 | 2.61 | 3.54 |
| | Speeding | 29.41 | 22 | 19.54 | 12.97 | 15.66 | 10.87 | 14.18 |
| | Red light violation | 0.52 | 0.71 | 0.27 | 0.2 | 0.23 | 0.3 | 0.33 |
| | Not stopping at STOP sign | 0.24 | 0.14 | 0.18 | 0.36 | 0.03 | 0.24 | 0.03 |
| | Drive opposite direction | 0.84 | 0.15 | 0.16 | 0.11 | 0.02 | 0.1 | 0.44 |
| | Illegal overtaking | 0.09 | 0.44 | 0.26 | 0.83 | 0.19 | 0.76 | 0.45 |
| | Not giving way | 11.7 | 15.17 | 1.44 | 16.64 | 16.76 | 18.23 | 17.89 |
| | Drifting | 0.14 | 0.59 | 0.19 | 0.14 | 0.21 | 0.14 | 0.03 |
| | Sudden lane change | 33.71 | 40.4 | 47.29 | 54.63 | 53.55 | 54.63 | 54.48 |
| | Insufficient safe distance | 7.57 | 7.84 | 4.3 | 6.13 | 6.03 | 8.24 | 4.15 |
| | Other | 2.03 | 3.49 | 4.97 | 4.48 | 3.56 | 3.86 | 4.49 |
| Vehicle Condition | Faulty tires | 99.53 | 99.39 | 99.71 | 99.55 | 99.62 | 99.73 | 99.58 |
| | Faulty brakes | 0.28 | 0.29 | 0.14 | 0.28 | 0.2 | 0.09 | 0.07 |
| | Faulty electric connection | 0.19 | 0.32 | 0.15 | 0.17 | 0.18 | 0.18 | 0.35 |
| Season | Winter | 25.59 | 22.91 | 26.55 | 25.33 | 25.94 | 23.53 | 24.55 |
| | Spring | 29.53 | 29.94 | 28.25 | 26.12 | 25.39 | 27.38 | 28.84 |
| | Summer | 24.78 | 27.98 | 24.35 | 26.66 | 23.46 | 24.4 | 23.12 |
| | Autumn | 20.09 | 19.17 | 20.84 | 21.89 | 25.21 | 24.68 | 23.48 |

### 5.2.1. Driver Education, Culture, and Awareness

Adults as well as teenagers are all exposed to and absorb traffic culture in different ways. This is accomplished through diverse visual, aural, and written media as well as suitable traffic awareness programs in schools, universities, institutes, and training facilities. It should be mentioned that increasing the level of driving schools and refining their curricula as well as emphasizing the need for traffic awareness are both critical components of this process. This would be especially beneficial to improving the behavior of young drivers, as aggressive driving behavior among young drivers is one of the most common causes of road accidents in Saudi Arabia [57]. A positive role model for children is provided by their father when he is driving his car, and traffic officers serve as an example because they spread awareness. It is therefore necessary to improve methods for selecting traffic system implementers and enrolling them in appropriate training in the field of traffic management.

### 5.2.2. Application of Modern Technologies

Connected and autonomous vehicles are poised to revolutionize mobility and transportation by displacing people as drivers and service providers. While the primary purpose of automated vehicles is to improve road safety and comfort, it also provides a tremendous chance to strengthen vehicle efficiency and to minimize emissions in the automotive industry. However, advances in automotive efficiency and usefulness are not always accompanied by net environmental sustainability gains [76].

Drivers who use a navigation system do not need to plan their own route, which results in less stress and more confidence behind the wheel. According to those who think that driving with a navigation system that delivers traffic information improves the quality of the chosen route, this has a positive impact on traffic safety. As a result, the journey time is reduced, and navigation errors are reduced [77,78]. There is information that navigation systems can assist drivers in navigating more effectively and simply than traditional approaches, such as by memorizing routes or utilizing paper maps. Eby and Kostyniuk reported that route guidance using in-car electronic aids leads to faster routes than written directions [79].

Based on the Intelligent Speed Adaptation System (ISAS) study, the Dutch Ministry of Transport estimates that if speed limits are strictly followed, road fatalities will be reduced by 20%, and hospitalized injuries will be decreased by 15%. Both fuel usage and carbon dioxide emissions will be lowered by 11% [80].

### 5.2.3. Legislation and Enforcement of Traffic Regulations

When an integrated traffic system is in place, it establishes fundamental rules that contribute to reducing traffic accidents and achieving the necessary security and goals of the political and social systems through the organization of traffic facilities; however, to achieve these goals, current systems must be reviewed and evaluated continuously, and work must be carried out to enact new regulations on issues that necessitate this action. Recently in Al-Ahsa, several modern steps have been taken to implement traffic safety regulations, including speed violation cameras, red light violation cameras, and mobile-usage and seat-belt violation cameras. It is expected that these steps will be able to improve road safety in Al-Ahsa.

### 5.2.4. Encourage Traffic Studies and Research

Supporting and encouraging research and scientific studies on the subject of traffic safety as well as fostering collaboration among all of the parties who are concerned with traffic safety, whether supervisory, executive, or academic, will be beneficial to improving road safety. To accomplish this, it is required to (i) pay close attention to traffic data and statistics and to develop an advanced information center dedicated to collecting, documenting, and monitoring all transportation and traffic data and statistics; additionally,

(ii) a portion of the budgets of insurance companies and revenues should be set aside for studies and associated research.

## 6. Conclusions

This study examines the causal links between several factors of driving behaviors related and road accidents in Al-Ahsa based on accident data collected over three years from 2015 to 2018. The causal links are represented as a model, which was constructed using the Bayesian belief network method. This model examines the relationships between the indicator variables while also estimating the uncertainty associated with accident outcomes. The BBN model was used to examine the sorts of accidents and severity that are caused by both purposeful and inadvertent driving behavior. According to the findings of this study, there is an increase in the likelihood of a collision based solely on speeding behavior. The likelihood of an accident increased when both speeding and brake failure were taken into account, more than doubling the previous estimate. The results exposed that the BBN model can effectively investigate the intricate links between driving behavior and accident causes that are inherent in traffic accidents. A notable advantage of the technique employed in this study is that it allows the reasoning process to be carried out in both the forward and backward directions at the same time. However, the drawback of this research is that female driving data were not analyzed because female driving was not authorized throughout the study period but has been subsequently permitted; nonetheless, data on female driving should be included in the design framework to ensure its effective operation.

**Author Contributions:** Conceptualization, M.M.R. and M.K.I.; methodology and software, M.M.R. and M.K.I.; formal analysis and validation, M.M.R. and M.K.I., resources and data curation, M.K.I. and M.A.; writing—original draft preparation, M.M.R., M.K.I., M.A. and A.A.-S.; review and editing, M.M.R., A.A.-S. and M.A.; project administration and funding acquisition, M.M.R. All authors have read and agreed to the published version of the manuscript.

**Funding:** This work was financially supported by the Deanship of Scientific Research in the King Faisal University, Saudi Arabia, through the project number GRANT 112.

**Institutional Review Board Statement:** Not applicable.

**Informed Consent Statement:** Not applicable.

**Data Availability Statement:** The data that support the findings of this study are available from the corresponding author, M.M.R. (mrahman@kfu.edu.sa), upon reasonable request.

**Acknowledgments:** The authors acknowledge the support received from King Faisal University (KFU) for conducting this research. Thanks to Syed Masiur Rahman of King Fahd University of Petroleum & Minerals (KFUPM), for initial review and suggestions.

**Conflicts of Interest:** The authors declare no conflict of interests.

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
