# Peer review of "Towards Sustainable Road Safety in Saudi Arabia: Exploring Traffic Accident Causes Associated with Driving Behavior Using a Bayesian Belief Network"

_sustainability, doi:10.3390/su14106315_

Round 1
Reviewer 1 Report
I carefully read the paper which using Al-Ahsa city in Saudi Arabia's Eastern province as a case study and a Bayesian Belief Network (BBN) model was established by incorporating an Expectation-Maximization algorithm. The theoretic analysis is verified by the field data. The findings demonstrated that the BBN model was capable of efficiently investigating the complex linkages between drivers’ behavior and accident causes that are inherent in road accidents. Hence, some revisions should be made before the publication by the journal. Some suggestions are shown in the following:
- Authors reported that the proposed method is novel, but there is no comparison with the existing literature. A detailed comparative analysis must be required to support the novelty of the work. For each part of the analysis or results discussion throughout the manuscript, authors need to provide the references to support the interpretations (Even some included already). Also, are the findings consistent or different from that of previous studies.
- What is the difference between the implemented Bayesian Belief Network and Bayesian Network. I cannot find the contribution of the developed model.
- The quality of the equations and figures of the paper should be improved, for example, the equation 2. The figures of the results are different to understand, for example figure 8. The figure should be made more understandable.
- The discussion of the paper should be improved. The authors should present what they found. Compare with previous studies, the difference and similarity of their findings should be presented.
- The results exposed that the BBN model could effectively investigate the intricate links between driving behavior and accident causes that are inherent in traffic accidents. A notable advantage of the technique employed in this study is that it allows the reasoning process to be carried out in both the forward and backward directions at the same time. Did other model can achieve the goal? What is the difference between your model and other model? You should give a comparison.
- The female driving data were not analyzed because female driving was not authorized throughout the study period. Maybe you could find some data in other area to evaluate the proposed model.
Author Response
We would like to thank the reviewer for the comments on our manuscript titled "Towards Sustainable Road Safety in Saudi Arabia: Exploring Traffic Accident Causes Associated with Driving Behavior Using a Bayesian Belief Network”, ID: Sustainability-1699887. We have revised the manuscript carefully based on the constructive comments and suggestions given by you. A complete list of the revisions in the revised version of the manuscript is given in the attached file. We would like to thank you for your support on improving both the content and the structure of the paper.
Regards,
Muhammad Muhitur Rahman, Md Kamrul Islam, Ammar Al-shayeb and Md Arifuzzaman

Reviewer 2 Report
The paper is certainly interesting for its attempt to individuate a scientific method to associate traffic accidents with driving behavior. All written is clear and readable but there is something to explain to better understand what written.
- Among accidents causes, authors include “not using seat belts” (page 2). In the model it seems not to be. The effect on accident of this cause is important, above all on accident severity. Why do not they consider it?
- About the discretized variables in the BBN, there are “seasons” whom distribution is quite constant, showing the fact that in Saudi Arabia the differences in seasons are not very low. Why do they choose these as variables?
- Page 17 of 20: which the link between the research results and outcomes and Drivers’ Education, culture, and awareness - Legislation and enforcement of traffic regulations - Encourage Traffic studies & research. It seems an introduction or a preliminary discussion rather than a BBN model discussion.
- Take a look to page number that are not continuous
Author Response

(The authors gave the same response as above.)

Reviewer 3 Report
This paper uses BBN approach to analyze the accident type and the injury severity. The Introduction part looks well. However, I have several concerns about the text, data and method. The comments are presented below.
1. In the Literature review, it could be better if the classifications of driving behaviors are listed in a table.
2. The authors mentioned some analysis tools which used to model crash frequency or rate on page 3, such as Poisson and NB models. I suggest that some recent modeling approaches regarding the severity of crash injury and classification of speeder would be helpful as well. Please refer the following studies
#1 https://doi.org/10.1016/j.aap.2021.106183
#2 https://doi.org/10.1016/j.aap.2018.04.019
#3 https://doi.org/10.1016/j.jsr.2020.12.009
3. Accident severity in this study only contains Fatal and Injury. Do the authors consider the property damage only (PDO) accident? The PDO accidents make up the major part of the accidents and ought to be investigated generally. Please explain.
4. In Table 1, the sums of the frequency of Unintentional Driver behavior and Vehicle condition are merely 3 and 48, respectively? Moreover, the sum of the frequency of Seasons samples equals to 4,000? Why these sums are far different from the accident frequency (3,994)?
5. According to the BBN illustrated in Figure 5, it seems that the Accident Severity is merely determined by Accident Type, which form a “one-to-one” relationship. However, the cause of different injury severity is complex and affected by various reasons beyond accident type. For instance, driving speed and the use of seat-belt (or helmet) are also vital to the injury severity. Therefore, the authors are supposed to develop a more general BBN with “many-to-one” relationships when inferring the injury severity.
6. In the end of this paper, the authors mentioned some improvements based on drivers’ education/culture/awareness, legislation and enforcement. However, since the topic of the current paper highlights the sustainable road safety, vehicular safety techniques ought to be discussed as well, such as automated vehicles, connective vehicles, intelligent speed adaptation system, advanced navigation system, etc.
7. The influence of intentional and unintentional driving behaviors on crash type and/or severity may change over time. This is not investigated by the current study but should be mentioned as a future research direction. The varying effect can be accounted for by the dynamic modeling approach, which can be found in the following references: https://doi.org/10.1016/j.amar.2021.100204; https://doi.org/10.1016/j.trc.2018.09.020; https://doi.org/10.1016/j.trc.2010.10.005
8. The authors are suggested to polish the paper in English writing.
Author Response

(The authors gave the same response as above.)

Round 2
Reviewer 1 Report
It can be published.
Reviewer 3 Report
All the reviewer comments have been carefully addressed.